DLX1 acts as a novel prognostic biomarker involved in immune cell infiltration and tumor progression in lung adenocarcinoma

Du Yu
Li Heng
Wang Yan
He Yunyan
Li Gaofeng 13987123539@126.com ligaofeng@kmmu.edu.cn
1 School of Clinical Oncology, Kunming Medical University , Kunming , China
2 Department of Thoracic Surgery, The Third Affiliated Hospital of Kunming Medical University , Kunming , China
Pfeffer Ulrich
Electronic publication date: 2024 Feb 2
Publication date: 2024
Volume: 12
Electronic Location ID: e16823
Received 2023 Jun 15; Accepted 2024 Jan 2
Copyright: ©2024 Du et al.
Copyright year: 2024
Copyright holder: Du et al.
License: This is an open access article distributed under the terms of the Creative Commons Attribution License, which permits unrestricted use, distribution, reproduction and adaptation in any medium and for any purpose provided that it is properly attributed. For attribution, the original author(s), title, publication source (PeerJ) and either DOI or URL of the article must be cited.
License URL: https://creativecommons.org/licenses/by/4.0/

Keywords: Prognosis, DLX1, Immune infiltrate, Lung adenocarcinoma

Funding: The High-level Medical and Health Professionals Training Fund of Yunnan Province L-2019028 The “Famous Doctor” Special Fund of Yunnan 10,000 People Plan CZ0096 This project is supported by the High-level Medical and Health Professionals Training Fund of Yunnan Province (L-2019028) and the “Famous Doctor” Special Fund of Yunnan 10,000 People Plan (CZ0096) from The Department of Thoracic Surgery of the Third Affiliated Hospital of Kunming Medical University. The funders had no role in study design, data collection and analysis, decision to publish, or preparation of the manuscript.

==============================
Background

The biological function of distal-less homeobox 1 (DLX1) in lung adenocarcinoma (LUAD) remains unclear, despite a growing body of evidence that DLX1 is involved in the initiation and progression of various tumors.

Methods

This study explored and confirmed the prognostic and immunologic roles of DLX1 in LUAD via bioinformatic analysis and cellular functional validation. MethSurv was used to analyze the DNA methylation levels of DLX1 and the prognostic value of CpG islands. DLX1 mutation rates and prognoses between patients with and without the mutated DLX1 gene were analyzed by cBioPortal. Finally, cellular functional assays were used to investigate the effect of DLX1 on LUAD cells.

Results

Our results showed that DLX1 mRNA expression was significantly upregulated in LUAD. High DLX1 expression or promoter methylation was associated with worse prognosis, which confirmed DLX1 as an independent prognostic factor in LUAD. The level of multiple immune cell infiltration was significantly associated with DLX1 expression. Genes in the high DLX1 expression group were mainly enriched in cell cycle checkpoint, DNA replication, DNA repair, Fceri-mediated MAPK activation, TP53 activity regulation, and MET activation of PTK2-regulated signaling pathways. Cellular functional assays showed that the knockdown of DLX1 inhibited the proliferation, migration, and invasion of LUAD cells.

Conclusion

Our study identified DLX1 as a potential diagnostic and prognostic biomarker, and a promising therapeutic target in LUAD.

Introduction

Lung cancer is one of the most commonly diagnosed cancers and the leading cause of cancer-related deaths worldwide (Thai et al., 2021). In 2020, there were an estimated 19.3 million new cancer cases and nearly 10 million cancer deaths worldwide. There are an estimated 2 million new lung cancer cases and an estimated 1.8 million deaths each year. The global burden of cancer is projected to reach 28.4 million cases in 2040, which is a 47% increase compared to 2020 (Sung et al., 2021). Early-stage disease (stage I/II) is asymptomatic and can be treated with curative intent. However, approximately 70% of lung cancer cases are diagnosed in advanced stages (III/IV) when treatment is rarely curative and the five-year survival rate is only 16.2%, which is a major contributor to the poor prognosis of lung cancer (Dickson et al., 2022). In contrast, the most common type of lung cancer is lung adenocarcinoma (LUAD), which is one of the most aggressive and fastest metastatic type of lung cancer, accounting for approximately 40% of lung cancer incidence (Denisenko, Budkevich & Zhivotovsky, 2018). Therefore, there is a growing need for novel biomarkers and therapeutic targets to improve the survival of LUAD patients.

Distal-less homeobox 1 (DLX1) belongs to the distal-less homeobox gene family, and DLX proteins are transcription factors homologous to Drosophila distal oligogenes, including six identified family members (DLX1-6). Three functionally related clusters, DLX1/DLX2, DLX3/DLX4, and DLX5/DLX6, overlap at specific sites on homologous chromosomes (Kraus & Lufkin, 2006). Initially, the DLX transcription factor (TF) was described as a master regulator of the developing vertebrate brain, driving forebrain gabaergic neuronal differentiation (Lindtner et al., 2019). Ablation of DlX1 and 2 altered the expression of genes that are critical for gabaergic development in the forebrain and involved in embryonic development, cell cycle regulation, apoptosis, and other processes (Tan & Testa, 2021). The association of DLX1 with the development of malignant tumors has been confirmed in several studies. Chan et al. (2017) found that DLX1 promotes ovarian cancer aggressiveness by activating the TGF-β/SMAD4 signaling pathway. Liang et al. (2018) found that DLX1 is a β-catenin binding protein, and aberrant activation of β-catenin/TCF signaling promotes the growth, migration, and invasion of prostate cancer cells. Furthermore, downregulation of the β-catenin/TCF signaling pathway inhibited the malignant development of prostate cancer cells (Chen et al., 2017). However, the relationship between DLX1 expression, tumor prognosis, and immune infiltration has remained unclear.

This study analyzed DLX1 expression levels in various cancers, including LUAD, from the Cancer Genome Atlas (TCGA) database. To identify differentially expressed genes (DEGs) between LUAD tissues with high and low DLX1 expression in the TCGA-LUAD database, RNA sequencing data were analyzed using the R software package. The relationship between DLX1 expression and immune cell infiltration was determined using Spearman correlation analysis. Logistic regression was used to analyze the relationship between DLX1 expression and clinicopathologic characteristics. The diagnostic and prognostic value of DLX1 was evaluated by Kaplan–Meier (K-M) survival curves, receiver operating characteristic (ROC) curves, nomogram models, and Cox regression analysis. The biological functions of DLX1 were determined using gene ontology (GO) and Kyoto Encyclopedia of Gene and Genomics (KEGG). MethSurv was used to analyze the DNA methylation levels of DLX1 and the prognostic value of CpG islands. The mutation rate of the DLX1 gene and the prognosis between patients with and without mutations in the gene were analyzed using cBioPortal. Finally, the biological function of DLX1 in LUAD was determined by cellular functional assays. Our data provide a basis for understanding the prognostic and immune infiltration-related role of DLX1 in LUAD.

Materials and Methods

TIMER database analysis

Tumor Immune Estimation Resource (TIMER, http://timer.cistrome.org/) is an online database for the comprehensive analysis of tumor-infiltrating immune cells and differential gene expression levels in various cancer types (Li et al., 2020). We used the TIMER database to investigate the differential expression of DLX1 in tumors and normal tissues in various cancer types.

RNA-sequencing data collection and bioinformatic analysis

We downloaded RNAseq data in level 3 HTSeq-FPKM format from 535 LUAD tumor tissues and 59 normal tissues from the TCGA (https://portal.gdc.cancer.gov/) LUAD project, as well as the corresponding clinicopathologic information. Data were transformed accordingly: fragments per kilobase per million (FPKM) RNAseq data were log2 transformed, and FPKM RNA sequencing data were converted to transcripts per million reads (TPM). The TCGA database is publicly available and written informed consent was obtained from patients prior to data collection. The 535 LUAD samples were divided into high and low DLX1 expression groups according to median DLX1 expression. DESeq2 was used with the absolute value of log-fold change > 1.5 and p-value < 0.05 as threshold parameters (Love, Huber & Anders, 2014). To analyze the DEGs between the two groups, we used the “ggplot2” software package to display the volcano map and heat map.

Functional enrichment analysis of DLX1-related DEGs in LUAD

Functional annotation and gene set enrichment analysis (GSEA) of DEGs were performed using the ClusterProfiler software package (Yu et al., 2012). Using absolute log fold change >1.5 and p < 0.05 as threshold parameters, we found 223 DEGs. The GSEA software was run using the MSigDB c2 collection (c2.all.v7.2.symbols.gmt). The functional enrichment analysis of DLX1-related DEGs in LUAD was visualized.

Kaplan–Meier plotter pan-cancer RNA-seq database analysis

The Kaplan–Meier mapper can be used to assess the prognostic impact of different genes in different types of cancer (http://kmplot.com). We performed the survival analysis of DLX1 expression in LUAD, as well as the prognostic value of DLX1 according to immune cells.

Correlation analysis of DLX1 expression level and immune cell infiltration in LUAD

The ssGSEA algorithm in GSVA (v1.34.0) R package (Hänzelmann, Castelo & Guinney, 2013) was used to evaluate the tumor penetration status of 24 kinds of immune cells (Bindea et al., 2013), including activated dendritic cells (aDC), B cells, CD8 T cells, cytotoxic cells, dendritic cells (DC), eosinophils, immature DC (iDC), macrophages, mast cells, neutrophils, CD56bright natural killer (NK) cells, CD56dim NK cells, NK cells, plasmacytoid DC (pDC), T cells, T helper cells, T central memory (Tcm), T effector memory (Tem) cells, T follicular helper (TFH) cells, T gamma delta (Tgd) cells, Th1 cells, Th17 cells, Th2 cells, and Treg cells. The relationship between DLX1 expression levels and immune cell infiltration status was determined using Spearman correlation analysis.

DLX1 in CpG island DNA methylation status analysis

The methylation status of the DLX1 gene CpG locus in the TCGA-LUAD dataset was analyzed using MethSurv (https://biit.cs.ut.ee/methsurv/). Additionally, the relationship between DLX1 gene CpG methylation status and LUAD overall survival (OS) was evaluated.

Genetic changes in LUAD samples

The following four LUAD datasets were analyzed in cBioPortal for genomic alterations in the DLX1 gene (https://www.cbioportal.org/): Broad, cell 2012; MSK, JThorac Oncol 2020; OncoSG, Nat Genet 2020; and TCGA, Firehose Legacy. K-M survival curve analysis and log-rank test were used to determine the prognostic significance of DLX1 mutations.

Correlation analysis between expression level of DLX1 and clinicopathologic features in LUAD patients

We extracted clinicopathologic data from the TCGA-LUAD project and previously published studies in LUAD patients, including overall survival (OS), disease-specific survival (DSS), and progression-free interval (PFI) (Liu et al., 2018). Software analysis was used to compare the differences in various clinicopathologic parameters between the high and low DLX1 expression groups, such as T stage, N staging, M staging, DSS events, OS events, PFI events, and pathologic staging. We used a normality test (p < 0.05) for normally distributed data, and Kruskal–Wallis test and Dunn’s multiple methods to analyze intergroup differences. The Bonferroni method was used to correct the significance level results. The “ggplot2” software package was used for statistical visualization. Logistic regression analysis was used to evaluate the relationship between DLX1 expression levels and the clinicopathologic characteristics of LUAD patients.

Prognostic significance of DLX1 expression in LUAD

Survival data of LUAD patients from the TCGA-LUAD project and previously published data (Liu et al., 2018) were analyzed using the “Survival” package (statistical analysis of survival data) and the “Survminer” package (visualization). Prognostic analyses were performed. To determine LUAD survival according to DLX1 expression level, we performed K-M survival curve analysis and univariate and multivariate COX regression analyses. Diagnostic ROC curves and nomogram model analysis were performed using “proc,” “time ROC” (statistical analysis), and “ggplot2” (visualization) software packages to evaluate the predictive value of DLX1 expression level in LUAD diagnosis. The K-M survival curve was used for prognostic analysis of a subset of LUAD patients. Results included pooled sample sizes (percent), hazard ratios (HRs), confidence intervals (CIs), and P values. Forest plots were generated using the “ggplot2” software package.

Cell lines, cell culture, and siRNA transfection

A549, HCC-827, H1299, and SPC-A-1 were obtained from the Shanghai Institutes for Biological Sciences, China, and the BEAS-2B lung epithelium cell line was obtained from the cell bank of the Kunming Animal Research Institute. All LUAD cell lines were grown in RPMI-1640 medium (Corning, Corning, NY, USA) supplemented with 10% fetal bovine serum (FBS) and 1% penicillin and streptomycin. BEGM medium (CC-3170, Lonza, Basel, Switzerland) was used to grow the BEAS-2B lung epithelial cell line. All cell lines were grown at 37 °C in humid air. H1299 and A549 cells were seeded in six-well plates 24 h before transfection. When the cell growth area was approximately 70%, cells were transfected with siRNA targeting DLX1 and negative control siRNA using Lipofectamine RNAimax reagent (Invitrogen, Waltham, MA, USA) according to the manufacturer’s protocol. The efficiency of the transfection was assessed by quantitative real-time RT-PCR (qRT-PCR) analysis.

Cell proliferation and cell migration assay

After cells were transfected with siRNA to target specific genes and negative control siRNA for 24 h, cells were harvested and counted in a cell counting chamber. Cells were then plated on 96-well plates at a density of 2,000 cells per 100 ml well. Cell proliferation was detected using the Cell Counting Kit-8. The optical density (OD) was measured at 450 nm every 24 h and for four consecutive days after the addition of CCK8 solution according to the manufacturer’s instructions. LUAD cells transfected with si-DLX1 and si-NC were cultured in six-well plates for 24 h. The surface of the cell layer was scratched with a pipette tip. Images were taken with a microscope 0 h and 24 h after damage. The distance of the damaged area at 24 h was measured. The relative mobility was calculated and evaluated by comparing the distance of the damaged area at 0 h. Cell migration and invasion were detected using Transwell chambers (Corning). To perform cell migration assays, 2 × 105 cells were added to the upper chamber and resuspended with 100 µL of serum-free RPMI 1640 medium, and 600 µL of RPMI 1640 medium containing 10% FBS was added to the lower chamber. After 24 h, the cells on the bottom surface of the lower chamber membrane were fixed with 4% paraformaldehyde and stained with 0.5% crystal violet, and the staining in five random fields was calculated by counting the number of cells under the microscope. For cell invasion, Matrigel was added to the upper chamber 3 h in advance. The remaining steps were performed as described above.

Real-time RT-PCR assay

The cells were lysed by RNA iso Plus (Takara Bio, Shiga, Japan) using real-time RT-PCR. This research use primer was as follows:

β-actin-F: GACCTGACTGACTACCTCATGAAGATCGC,

β-actin-R: GTCACACTTCATGATGGAGTTGAAGG,

DLX1-F: CGCTTCAATGGCAAGGGAAA,

DLX1-R: CTCTCCGGCAGAGCTAGGTA.

Statistical analysis

DLX1 expression in the normal and LUAD groups was statistically analyzed using the Wilcoxon rank sum test. We divided the patients into two groups according to the median expression of DLX1. Clinicopathological characteristics of DLX1 were analyzed using the Wilcoxon rank sum test, Kruskal–Wallis test, and logistic regression. Prognostic analysis was performed using Kaplan–Meier survival analysis and Cox univariate and multifactorial analysis. To evaluate the diagnostic significance of DEGs, ROC curves were plotted using the “plotROC” program package. The “RMS” R program package was used to plot the prognostic value of LUAD patients on the nomogram. All statistical analyses were performed in the R environment (v3.6.3). Wound healing and Transwell experiments were analyzed using Image J software. We used GraphPad Prism 8.0 software to graph the data, Student’s t-test to determine the significance of data between two experimental groups, and one-way ANOVA for multiple group comparisons. Each experiment was repeated at least three times and expressed as mean ± standard deviation (mean ± SD). P values < 0.05 were considered statistically significant differences (*: p < 0.05, **: p < 0.01, ***: p < 0.001).

Results

DLX1 expression levels are significantly elevated in multiple cancers including LUAD

When analyzing the expression of DLX1 in different tumor types online in TIMER (https://cistrome.shinyapps.io/timer/), we found that the expression of DLX1 was significantly higher in malignant tissues than in normal tissues (p < 0.05, Fig. 1A). DLX1 expression in different malignant tumor types was analyzed using the TCGA database, followed by paired difference analysis. The results showed that DLX1 expression was significantly higher in tumor tissues compared to normal tissues (p < 0.05, Fig. 1C). We also determined that DLX1 expression in pan-cancer paired cancer tissues and adjacent normal tissues by means of the TCGA dataset. DLX1 levels were significantly higher in 12 of the 18 cancers compared to normal tissues (p < 0.05, Fig. 1B). DLX1 was significantly more expressed in unpaired samples (Fig. 1C) versus paired samples (Fig. 1B) in LUAD than in normal tissues (p <0.05).

Figure 1 DLX1 expression was significantly upregulated in several tumors, including lung adenocarcinoma.

(A) DLX1 expression in different cancer types from the TIMER database. (B) Expression of DLX1 in matched and adjacent normal tissues from the TCGA database. (C) DLX1 expression was significantly elevated in LUAD compared to normal tissues in the TCGA-LUAD dataset. Ns: * p < 0.05, ** p < 0.01, *** p < 0.001.

The expression of DLX1 in LUAD was up-regulated, which indicated a poor prognosis in patients with LUAD

DLX1 expression and clinical significance were analyzed in the TCGA database. We found that OS, DSS, and PFI were shorter in LUAD patients with higher DLX1 levels (Figs. 2A–2C). We further investigated the diagnostic value of DLX1 in the differentiation of LUAD samples from normal lung tissue. The ROC curve analysis confirmed that the AUC value of DLX1 was 0.754, 95% CI [0.703–0.805] (Fig. 2D). These results suggest that DLX1 is upregulated in LUAD and that high expression levels of DLX1 are associated with poor prognosis in lung cancer patients. Based on the validation of the prognostic value of DLX1 in different subgroups, we further determined the prognostic value of DLX1 in different clinical subgroups, namely pathological stage, tumor lymph node metastasis (TNM) stage, gender, age, race, smoking status, and smoking age. The results showed that the upregulation of DLX1 levels was associated with poor clinical outcomes in lung cancer patients (Figs. 2E–2G).

Figure 2 Kaplan–Meier curve of DLX1 and ROC curve and total survival of DLX1 based on different subpopulations.

(A–C) Kaplan–Meier survival curve of the TCGA-LUAD dataset showed poor OS, DSS, and PFI of DLX1 in LUAD patients with high DLX1 expression. (D) Diagnostic value of DLX1 in lung adenocarcinoma was determined using ROC curves. (E–G) Association of DLX1 expression levels with overall survival in different LUAD clinical subgroups including stage I and II, stage T1 and T2, stage N0, stage M0, smoker status, sex, age, white, and smoking age.

Relationship between the expression of DLX1 and clinical characteristics of LUAD

We found that DLX1 expression was significantly correlated with pathological stage, OS events, DSS events, and PFI events in LUAD patients (Table 1). In addition, logistic analysis showed that DLX1 upregulation correlated with T stage (T2 & T3 & T4 vs. T1) and pathological stage (stage II, stage III, and stage IV vs. stage I) (Table 2).

Table 1 Clinicopathological features of LUAD patients with high and low DLX1 expression.

Bold numbers represent significant values.

Characteristic	Low expression of DLX1	High expression of DLX1	p	
n	256	257		
T stage, n (%)			0.237	
T1	94 (18.4%)	74 (14.5%)		
T2	132 (25.9%)	144 (28.2%)		
T3	23 (4.5%)	24 (4.7%)		
T4	7 (1.4%)	12 (2.4%)		
N stage, n (%)			0.050	
N0	169 (33.7%)	161 (32.1%)		
N1	37 (7.4%)	58 (11.6%)		
N2	43 (8.6%)	31 (6.2%)		
N3	1 (0.2%)	1 (0.2%)		
M stage, n (%)			0.354	
M0	178 (48.2%)	166 (45%)		
M1	10 (2.7%)	15 (4.1%)		
Pathologic stage, n (%)			0.019	
Stage I	149 (29.5%)	125 (24.8%)		
Stage II	47 (9.3%)	74 (14.7%)		
Stage III	45 (8.9%)	39 (7.7%)		
Stage IV	10 (2%)	16 (3.2%)		
OS event, n (%)			0.030	
Alive	175 (34.1%)	151 (29.4%)		
Dead	81 (15.8%)	106 (20.7%)		
DSS event, n (%)			0.007	
Alive	196 (41.1%)	166 (34.8%)		
Dead	45 (9.4%)	70 (14.7%)		
PFI event, n (%)			0.034	
Alive	164 (32%)	140 (27.3%)		
Dead	92 (17.9%)	117 (22.8%)		
Gender, n (%)			0.504	
Female	142 (27.7%)	134 (26.1%)		
Male	114 (22.2%)	123 (24%)		
Age, median (IQR)	66 (59, 73)	66 (59.5, 72)	0.774	

Table 2 Logistic regression analyzed the correlation between DLX1 expression and clinicopathologic features of LUAD.

Bold numbers represent significant values.

Characteristics	Total (N)	Odds Ratio (OR)	P value	
T stage (T1 vs. T2 & T3 & T4)	532	1.465 (1.019–2.113)	0.040	
N stage (N0 vs. N1 & N2&N3)	519	1.150 (0.797–1.661)	0.454	
M stage (M0 vs. M1)	386	1.594 (0.705–3.755)	0.269	
Pathologic stage (Stage I vs. Stage II & Stage III & Stage IV)	527	1.487 (1.053–2.103)	0.025	
Gender (Male vs. Female)	535	1.137 (0.809–1.598)	0.460	
Age (>65 vs.≤ 65)	516	1.047 (0.741–1.479)	0.795	
Smoker (Yes vs. No)	521	1.113 (0.682–1.821)	0.669	
Residual tumor (R0 vs. R1 & R2)	372	0.864 (0.318–2.310)	0.769	

Functional enrichment analysis of DLX1-related DEGs in LUAD

Based on the median DLX1 expression, the 535 LUAD patients were divided into high DLX1 expression group and low DLX1 expression group. We then found 223 DEGs in the high DLX1 expression group compared to the low DLX1 expression group (184 up-regulated and 39 down-regulated) using absolute log fold change >1.5 and p < 0.05 as threshold parameters (Fig. 3A). The single gene co-expression heat map in Fig. 3B shows the top 100 DEGs (Fig. 3B).We then performed functional annotation of DLX1-associated DEGs in LUAD patients using the “clusterProfiler” R package. The results of GO and KEGG enrichment analysis included highly enriched biological process (BP), cellular competent (CC), and molecular function (MF) (p < 0.05). The most common biological processes included “multicellular biological processes,” “apoptotic processes involved in development,” and “apoptotic processes involved in development” (p < 0.05). The most common biological processes included “multicellular biological processes,” “apoptotic processes involved in development,” “positive regulation of cell secretion,” and “γ-aminobutyric acid signaling pathway.” The most abundant cellular components were “GABA receptor complex,” “synaptic membrane,” “ion channel transporter complex,“ and “transport vesicles.” The most abundant cellular components were “GABA receptor complex,” “synaptic membrane,” “ion channel transporter complex,” “transport vesicles,” “cell base, and “intermediate filaments”. The most abundant molecular functions were “channel activity,” “motor activity,” “neurotransmitter receptor activity,” and “DNA-binding transcriptional repressor activity.” The most abundant molecular functions were “channel activity,” “motor activity,” “neurotransmitter receptor activity,” and “DNA binding transcriptional repressor activity.” The most common metabolic pathways include “ascorbate and aldehyde metabolism,” “chemical carcinogenesis,” “steroid hormone biosynthesis,” “neuroactive ligand–receptor interaction,” and “neurotransmitter receptor activity” (Figs. 3C–3F). GSEA revealed that the DEGs associated with DLX1 were significantly enriched in a cluster of cell proliferation-related genes (Fig. 4), including genes associated with cell cycle checkpoints (Nes = 2.012, Padj = 0. 036, FDR = 0.029), G2-M checkpoints (Nes = 1.780, Padj = 0.036, FDR = 0.029), M phase (Nes = 1.952, Padj = 0.036, FDR = 0.029), S phase (Nes = 1.589, Padj = 0.047, FDR = 0.037), mitotic G2-M phase (Nes = 1.867, Padj = 0.036, FDR = 0. 029), DNA repair (Nes =1.725, Padj = 0.036, FDR = 0.029), DNA replication (Nes = 1.752, Padj = 0.036, FDR = 0.029), gene expression regulation (Nes = 1.623, Padj = 0.042, FDR = 0.034), and DNA strand extension (Nes = 1.899, Padj = 0. 036, FDR = 0.029.) DLX1-associated DEGS were significantly associated with Fceri-mediated MAPK activation regulated by TP53 activity and MET activation of PTK2-regulated signaling pathways (Table S2).

Figure 3 Functional enrichment analysis.

(A) Volcano map and (B) heat map showed correlation analysis of DLX1 expression and its top 100 co-expressed gene networks. (C–F) GO and KEGG enrichment analysis revealed biological process (BP), cellular component (CC), and molecular function (MF), as well as KEGG terminology co-expressed with DLX1.

Figure 4 Study of DLX1-associated signaling pathways in lung adenocarcinoma.

(A–D) GSEA software identifies DLX1-associated signaling pathways.

The expression level of DLX1 is related to the infiltration of various immune cells in LUAD

ssGSEA was used to evaluate the infiltration of 24 types of immune cells in LUAD tissues. To evaluate the correlation between immune cell infiltration and expression, the association with DLX1 was evaluated using Spearman correlation analysis (Fig. 5A).The expression level of DLXI was positively correlated with Th2 cells (R = 0.167) (Fig. 6B) and negatively correlated with B cells (R = 0.091) (Fig. 5B), CD8 T cells (R = −0.137) (Fig. 5C), cytotoxic cells (R = −0.118)) (Fig. 5D), eosinophils (R = −0.149) (Fig. 5E), iDC (R = −0.089) (Fig. 5F), DC (R = −0.086) (Fig. 5G), Th17 cells (R = −0.132) (Fig. 6C), T cells (R = −0.110) (Fig. 6D), TFH (R =−0.094) (Fig. 6E), Mast cells (R = 0.157) (Fig. 6F), and neutrophils (R = −0.133) (Fig. 6G). The level of immune cell infiltration in the tumor was consistent with the results of Spearman analysis (Fig. 6A). Analysis of OS in immune cell infiltration DLX1 expression by Kaplan–Meier mapping (http://kmplot.com) showed poor DLX1 survival in high expressing type 2 helper T cells and poor DLX1 survival in low expressing B cells, CD4T cells, and eosinophils (Fig. 7).

Figure 5 Correlation between DLX1 expression and immune cell infiltration levels in LUAD.

(A) sGSEA analysis for correlation between DLX1 expression and 24 immune cell infiltration levels. (B–G) DLX1 expression correlated with immune infiltration levels in (B) B cells, (C) CD8 T cells, (D) cytotoxic cells, (E) eosinophils cells, (F) iDC helper cells, and (G) DC cells. Ns: * p < 0.05, ** p < 0.01, *** p < 0.001.

Figure 6 Correlation between DLX1 expression and immune cell infiltration levels in LUAD.

(A) A grouping comparison of DLX1 expression and 24 immune cell infiltration levels. (B–G) DLX1 expression correlated with immune infiltration levels in (B) Th2 cells, (C) Th17 cells, (D) T cells, (E) TFH cells, (F) mast cells, and (G) neutrophil cells. Ns: * p < 0.05, ** p < 0.01, *** p < 0.001.

Figure 7 DLX1 has high prognostic value in LUAD patients.

Kaplan–Meier mapper database analysis revealed differences (A–H) between LUAD patients with high and low levels of DLX1 expression and immune cells. P < 0.05 was statistically significant.

Methylation status of DLX1 gene is associated with prognosis of patients with LUAD

DNA methylation levels in the DLX1 gene and the prognostic value of CpG islands in DLX1 gene were analyzed using the MethSurv tool. The results showed 32 methylated CpG islands, including cg07936950, cg05938001, cg12308746, and cg17737681, with elevated DNA methylation levels (Fig. 8). Furthermore, the methylation levels of two CpG islands (cg05938001 and cg06996609) were prognostic (p < 0.05) (Table 3). Higher DLX1 methylation in these two CpG islands, particularly cg05938001, was associated with worse OS in LUAD patients compared with LUAD patients with lower DLX1 CpG methylation.

Figure 8 DNA methylation levels in DLX1 genes are associated with prognosis in LUAD patients.

Table 3 Effect of CpG locus methylation level of DLX1 gene on prognosis in LUAD patients.

Bold numbers represent significant values.

Name	CpG island	HR for OS (95% CI)	p-value	
cg00822840	Body-Island	1.291 (0.892; 1.869)	0.1671	
cg01244270	3′UTR-S_Shore	1.072 (0.779; 1.476)	0.67109	
cg03141620	TSS200-Island	1.086 (0.764; 1.544)	0.64256	
cg03479694	3′UTR-Island	1.23 (0.899; 1.683)	0.19604	
cg05094081	Body-N_Shore	0.965 (0.705; 1.32)	0.82194	
cg05938001	Body;3′UTR-Island	1.508 (1.013; 2.246)	0.03522	
cg06996609	TSS200-Island	1.456 (0.998; 2.125)	0.04382	
cg07130890	TSS1500-Island	1.342 (0.924; 1.95)	0.11309	
cg07181565	3′UTR-S_Shore	1.135 (0.829; 1.553)	0.43059	
cg07936950	3′UTR-Island	0.822 (0.578; 1.168)	0.28111	
cg08530978	TSS200-Island	1.123 (0.788; 1.6)	0.518	
cg10709593	Body;3′UTR-Island	1.24 (0.906; 1.697)	0.17846	
cg10791926	TSS1500-Island	1.266 (0.925; 1.733)	0.1404	
cg11575738	TSS200-N_Shore	0.794 (0.559; 1.127)	0.20442	
cg12308746	Body-N_Shore	1.17 (0.812; 1.687)	0.39292	
cg12676702	3′UTR-S_Shore	1.096 (0.801; 1.501)	0.5666	
cg12762816	3′UTR-S_Shore	1.213 (0.882; 1.669)	0.23759	
cg13343818	Body-Island	1.289 (0.942; 1.763)	0.11362	
cg15236866	TSS1500-N_Shore	0.963 (0.702; 1.321)	0.81618	
cg15552158	3′UTR-S_Shore	1.23 (0.894; 1.692)	0.20712	
cg15941449	TSS1500-Island	0.805 (0.585; 1.106)	0.178	
cg16347858	Body-Island	1.077 (0.754; 1.537)	0.68194	
cg16652259	TSS1500-Island	1.047 (0.729; 1.504)	0.80368	
cg17737681	Body-Island	1.175 (0.82; 1.684)	0.37414	
cg17780624	TSS1500-Island	1.245 (0.86; 1.802)	0.23763	
cg18147215	Body-Island	1.364 (0.945; 1.967)	0.08922	
cg18619591	TSS200-Island	1.108 (0.81; 1.515)	0.52207	
cg21038382	5′UTR;1stExon-N_Shore	0.833 (0.583; 1.19)	0.30843	
cg22977876	TSS200-Island	0.945 (0.678; 1.316)	0.73514	
cg24228707	TSS1500-Island	1.149 (0.794; 1.663)	0.45522	
cg26067250	1stExon-N_Shore	0.941 (0.687; 1.29)	0.70611	
cg27378835	1stExon-N_Shore	0.808 (0.59; 1.107)	0.18356	

The gene change of DLX1 had no effect on the survival outcome of patients with LUAD

Genetic alterations in the DLX1 gene were then analyzed using samples from 1,678 LUAD patients from the following four datasets: Broad, cell 2012 (n = 183); MSK, J Thorac Oncol 2020 (n = 604); OncoSG, Nat Genet 2020 (n = 305); and TCGA, Firehose Legacy (n = 586). Genetic alterations in the DLX1 gene were observed in only 1.5% of LUAD patients (Fig. 9A). K-M survival curves showed no significant difference in OS (p = 0.209) and DSS (p = 0.444) in patients with or without DLX1 gene alterations (Figs. 9B, 9C).

Figure 9 DLX1 alterations were not associated with survival outcomes in LUAD.

(A) Oncoprint visual summary of DLX1 gene alterations. (B, C) Kaplan–Meier survival curves showed (B) overall survival and (C) disease-specific survival in LUAD patients with or without DLX1 gene alterations.

Univariate and multivariate Cox regression analysis of OS, DSS, and PFI with different parameters

We performed univariate Cox regression analysis in the TCGA-LAUD cohort to determine whether DLX1 expression level could be used as a valuable prognostic biomarker (Table 4). Univariable Cox regression results showed that high DLX1 expression, pathological stage, TNM stage, presence of residual lesions after treatment, and treatment outcome were associated with OS and DSS in LUAD patients (Figs. 10A–10B). To determine whether DLX1 expression level could be an independent prognostic factor for LUAD patients, multifactorial Cox regression analysis was performed. We confirmed that increased DLX1 expression was a significant independent prognostic factor in the TCGA-LUAD cohort, which was directly correlated with pathological stage, the presence of residual foci after treatment, and the outcome of treatment (Figs. 10A–10B). Cox regression analysis revealed that DLX1 expression, pathologic staging, T-stage, N-stage, presence of residual disease after therapy, and outcome were associated with PFI in patients with LUAD (Fig. 10C). To determine whether DLX1 expression level could be an independent prognostic factor for LUAD patients, multifactorial Cox regression analysis was performed. We found that DLX1 expression was a significant independent prognostic factor in the TCGA-LUAD cohort, with a direct correlation with pathologic stage, presence of post-treatment residual lesions, and response rate (Fig. 10C).

Table 4 Cox regression analysis of clinical outcomes in LUAD patients based on clinicopathologic features including DLX1 levels.

Bold numbers represent significant values. Footnotes *, ** and *** in Table 4: Ns: p > 0.05,*p ¡ 0.05, **p < 0.01, ***p < 0.001.

Characteristics	HR for overall survival (95% CI)	HR for disease specific survival (95% CI)	HR for progress-free Interval (95% CI)	
	Univariate	Multivariate	Univariate	Multivariate	Univariate	Multivariate	
T stage (T1/T2 vs. T3/T4)	1.728 ∗∗	1.273	1.85 ∗∗	1.237	1.882 ∗∗∗	1.342	
N stage (N0 vs. N1/N2/ N3)	2.601 ∗∗∗	1.411	2.703 ∗∗∗	1.429	1.512 ∗∗	0.932	
M stage (M0 vs. M1)	2.136 ∗∗	1.021	2.455 ∗∗	0.945	1.513		
Pathologic stage (Stage I & Stage II vs. Stage III & Stage IV)	2.664 ∗∗∗	2.097 ∗	2.436 ∗∗∗	2.314 ∗	1.513 ∗	1.046	
Primary therapy outcome (PD & SD & PR vs. CR)	0.372 ∗∗∗	0.318 ∗∗∗	0.25 ∗∗∗	0.225 ∗∗∗	0.273 ∗∗∗	0.247 ∗∗∗	
Gender (Female vs. Male)	1.07		0.989		1.172		
Age (≤65 vs. >65)	1.223		1.013		1.023		
Residual tumor (R0 vs. R1 & R2)	3.879 ∗∗∗	4.398 ∗∗∗	4.743 ∗∗∗	7.396 ∗∗∗	3.091 ∗∗∗	2.526 ∗	
Smoker (No vs. Yes)	0.894		1.04		0.968		
DLX1 (Low vs. High)	1.548 ∗∗	1.637 ∗	2.047 ∗∗∗	2.181 ∗∗	1.615 ∗∗∗	1.948 ∗∗∗	

Figure 10 Forest map for single-variable and multi-variable Cox regression analysis in LUAD.

(A) Overall survival (OS), (B) disease-specific survival (DSS), and (C) progression-free interval (PFI) in LUAD.

Construction and verification of normograph based on DLX1

DLX1 was found to be an independent prognostic factor for LUAD in a multifactorial analysis. By combining DLX1 expression with TNM stage, pathological stage, and presence of residual foci after treatment, we constructed predictive models for OS, DSS, and PFS. To incorporate DLX1 as a LUAD biomarker, we constructed a nomogram. Higher overall OS, PFS, and DSS scores were associated with poorer prognosis (Figs. 11A–11C).

Figure 11 Construction and performance validation of DLX1 standard maps in lung adenocarcinoma patients.

(A–C) Calibration curves and Hosmer-Lemesow assays for (A) overall survival, (B) disease-specific survival, and (C) progression-free survival in the TCGA-lung adenocarcinoma cohort.

Depletion of DLX1 significantly suppressed proliferation and migration of LUAD cells

To detect the expression of DLX1, we examined the expression level of DLX1 in LUAD cell lines using qRT-PCR. The results confirmed that DLX1 was significantly up-regulated in lung cancer cell lines, specifically H1299 and A549 cells (Fig. 12A). qRT-PCR analysis showed that DLX1 mRNA expression was significantly decreased after transfection of H1299 and A549 cells with si-DLX1 (Figs. 12B–12C). Cell counting kit-8 (CCK8), Transwell, and wound healing assay analysis showed that the deletion of DLX1 significantly inhibited the cell proliferation and the cell migration ability of LUAD (Figs. 12D–12K). These data indicate that DLX1 is highly expressed in LUAD and significantly affects its proliferation, migration, and invasion.

Figure 12 DLX1 regulates proliferation and migration of LUAD cells.

(A) qPCR detected DLX1 expression in lung adenocarcinoma cell lines H1299, A549, HCC827, and SPC-A1 compared to normal human bronchial epithelial cell lines BEAS-2B. (B, C) Establishment of DLX1 knockdown low cell lines (D–I) in A549 and H1299 was validated by real-time RT-PCR and low DLX1 knockdown significantly inhibited proliferation and migration in A549 and H1299 cells as measured by CCK8, Transwell, and wound healing assays. NC, negative control; siRNA, DLX1 siRNA. Ns: p > 0.05,* p < 0.05, ** p < 0.01, *** p < 0.001.

Discussion

Using bioinformatics analysis, we determined the expression of DLX1 in LUAD and its clinical significance. The expression of DLX1 in LUAD was significantly higher than in normal tissues, and high DLX1 expression was significantly associated with poorer prognosis in LUAD patients. Furthermore, DLX1 was closely associated with immune cell infiltration in LUAD. Finally, cellular functional assays demonstrated that DLX1 gene deletion significantly inhibited the proliferation, migration, and invasion of LUAD cells. This study provides important information for understanding the function of DLX1, which can be used as a prognostic marker for tumor progression and immune cell infiltration in LUAD.

In addition, DLX1 expression levels in LUAD tissue were significantly associated with clinical stage, OS, DSS, and PFI. Logistic regression analysis showed that DLX1 expression level correlated with T stage and pathological stage. K-M survival curves showed that OS, DSS, and PFI rates were significantly lower in patients with high DLX1 expression than in DLX1 controls with lower expression levels. We also created a nomogram to integrate DLX1 as a LUAD biomarker; higher total nomogram scores for OS, DSS, and PFS indicated worse prognosis.

Several studies have reported evidence for the role of the DLX gene family in the formation of multiple cancers and their potential as a biomarker for diagnosis and prognosis. For example, DLX2 leads to cellular carcinogenesis by reducing cellular senescence through the regulation of p53 function (Tan & Testa, 2021), while DLX4 is also associated with metastasis in breast cancer. DLX4 induces cancer cells to undergo epithelial-to-mesenchymal transition (EMT) via TWIST (Tan & Testa, 2021), and the overexpression of DLX4 increases TWIST expression in cancer cell lines, leading to enhanced migration and invasion (Zhang et al., 2012). DLX4 induces CD44 by stimulating IL-1β-mediated NF-κB activity, thereby promoting ovarian cancer metastasis. DLX5 promotes osteosarcoma progression by activating the NOTCH signaling pathway (Zhang et al., 2021). DLX6 promotes oral cancer cell proliferation and inhibits apoptosis, and the EGFR-CCND1 signaling pathway may be a potential mechanism involved in the regulatory axis (Liang et al., 2022). However, the functional role of DLX1 in LUAD has not been reported.

To further examine the functional role of DLX1 in LUAD, we enriched the DEGs of DLX1 with GO and KEGG enrichment analysis, which showed that these genes are involved in the most common biological processes including “multicellular biological processes,” “apoptotic processes involved in development,” “positive regulation of cell secretion,” and “signaling pathways.” The most common biological processes in which these genes are involved included “multicellular biological processes,” “apoptotic processes involved in development,” “positive regulation of cell secretion,” and “signaling pathways.” The most abundant cellular components were “GABA receptor complex,” “synaptic membrane,” “ion channel transporter complex,” and “transport vesicles.” The most abundant cellular components were “GABA receptor complex,” “synaptic membrane,” “ion channel transporter complex,” “transport vesicles,” “cell base, and “intermediate filaments”. The most abundant molecular functions were “channel activity,” “motor activity,” “neurotransmitter receptor activity,” and “DNA-binding transcriptional repressor activity.” The most abundant molecular functions were “channel activity,” “motor activity,” “neurotransmitter receptor activity,” and “DNA binding transcriptional repressor activity.” The most common metabolic pathways include “ascorbate and aldehyde metabolism,” “chemical carcinogenesis,” “steroid hormone biosynthesis,” “neuroactive ligand–receptor interaction,” and “neurotransmitter receptor activity.” Song et al. (2016) found that γ-aminobutyric acid (GABA) inhibits cell cycle progression through G2/M or G1/S phases and therefore plays an inhibitory role in tumor proliferation. In addition, Jiang et al. (2012) showed that GABA(B) receptor activation not only inhibited the proliferation and migration of various human tumor cells, but also led to inactivation of cAMP response element binding protein (CREB) and ERK in tumor cells, confirming its potential as a therapeutic target in human cancers. GSEA enrichment analysis showed that upregulation of DLX1 expression was associated with DNA repair, DNA replication, gene expression regulation, Fceri-mediated MAPK activation, regulation by TP53 activity, and MET activation of PTK2-regulated signaling pathways. Tong et al. (2019) showed that PTK2 inhibitors and EGFR-TKI restored EGFR-TKI sensitivity in EGFR-TKI-resistant NSCLC. This suggests that PTK2 inhibitors play an important role in patients with acquired EGFR-TKI resistance. There are four classical pathways of MAPK, and these signaling pathways are closely associated with the development of several cancer species (Hargadon, 2023; Imperial et al., 2019).

We decided to explore the biological function of DLX1 in LUAD. In vitro, we found that DLX1 expression increased in LUAD cell lines. Since DLX1 expression was relatively high in A549 and H1299, we selected A549 and H1299 cells for our experiments. We found that in A549 and H1299 cells, DLX1 deletion inhibited cell proliferation, migration, and invasion. Based on these results, we suggest that DLX1 plays an important role in regulating the pathological progression of LUAD.

Our study demonstrated that the relationship between DLX1 expression and tumor immune cell infiltration was mostly negative. DLX1 expression was negatively correlated with B cells, CD8T cells, cytotoxic cells, eosinophils, iDC, DC, Th17 cells, T cells, TFH, mast cells, and neutrophils. Neutrophils exert anti-tumor effects by activating immune responses against tumor cells and by directly lysing them (Linde et al., 2023). In contrast, DCs are precisely recognized by cancer antigens, cross-presented for CD8+ T cell initiation, and recognized and killed by sensitized CD8+ T cells. It is from these two immune recognition processes that the immune escape of tumors occurs (Jhunjhunwala, Hammer & Delamarre, 2021). Cytotoxic T cells, one of the effector cells responsible for tumor regression, play an important role in lung cancer immunity (Phimister & Rubin, 2022). DLXI expression levels were positively correlated with Th2 cells (R = 0.167, p < 0.001), which are responsible for anti-tumorigenesis mediated through tumor microenvironment reprogramming. Th2 cells are responsible for the production of apoptotic factors and affect the recruitment of macrophages as well as eosinophils into the tumor where they produce cytotoxic factors to directly kill tumor cells (Jacenik, Karagiannidis & Beswick, 2023). Our data suggest that DLX1 overexpression plays an important role in the immune escape mechanism of LUAD cells, which promotes the growth and progression of LUAD.

DNA methylation is a common epigenetic mechanism that plays an equally important role in tumorigenesis, as reported in several studies. An altered methylation status of some genes is associated with the initiation, growth, and progression of various cancers, and site-specific hypermethylation of the gene body canyon without DLX1, but not the promoter, can directly increase its gene expression (Su et al., 2018). We investigated the relationship between DLX1 gene methylation levels and prognosis in LUAD patients. Hypermethylation at two CpG sites (cg05938001 and cg06996609) was associated with poorer OS. cg07936950, cg05938001, cg12308746, and cg17737681 were the four CpG sites with the highest DNA methylation. Targeted therapy is also an important component of lung cancer treatment, and treatment regimens require testing for EGFR, ALK, ROS1, RET, MET, BRAF, NTRK, HER2, NRG1, FGFR1, and PIK3CA (Guo et al., 2019; Wei et al., 2023). When further testing the sensitivity of DLX1 gene mutations in LUAD, we found that the incidence of DLX1 gene mutations in LUAD tissues was only 1.5% by cBioPortal study. In addition, DLX1 gene mutation was not associated with OS and DSS in LUAD patients.

These data point to DLX1 as a potential diagnostic and prognostic biomarker for LUAD. However, our study had the following limitations: we did not investigate the downstream signaling pathways and relative protein levels of DLX1 in LUAD tissues, and our results were based only on RNA sequencing data from LUAD tissues in the TCGA database. Therefore, to fully investigate the mechanism of DLX1 in LUAD, further in vivo and ex vivo experiments are required.

Conclusion

In conclusion, DLX1 is upregulated in LUAD tissues compared to their matched adjacent normal tissues. High expression of DLX1 is closely correlated to the advanced pathological stage and poor prognosis. ROC analyses partially indicate that DLX1 is a good predictive biomarker for differentiating lung adenocarcinoma from normal tissue. Moreover, DLX1 is possibly affect the progression of LUAD by regulating the expression of cell cycle and immune response related genes. Further studies in vitro and in vivo are required to confirm our findings.

Supplemental Information

Supplemental Information 1 GO KEGG

Click here for additional data file.

Supplemental Information 2 GSEA

Click here for additional data file.

Supplemental Information 3 Wound healing assay

Click here for additional data file.

Supplemental Information 4 A549 wound healing assay

Click here for additional data file.

Supplemental Information 5 H1299 wound healing assay

Click here for additional data file.

Supplemental Information 6 WB

Click here for additional data file.

Supplemental Information 7 CCK8

Click here for additional data file.

Supplemental Information 8 qPCR

Click here for additional data file.

Supplemental Information 9 qPCR original data

Click here for additional data file.

Supplemental Information 10 qPCR result

Click here for additional data file.

Supplemental Information 11 Transwell

Click here for additional data file.

Supplemental Information 12 Heat map of expression pattern of long survival time of core genes

Click here for additional data file.

Supplemental Information 13 K-M survival curve

Click here for additional data file.

Supplemental Information 14 PCR experimental verification

Click here for additional data file.

Supplemental Information 15 DLX1 Antibody Basic Information

Click here for additional data file.

Supplemental Information 16 Molecular Weight of DLX1 Antibody Protein

Click here for additional data file.

The authors would like to thank the Yunnan Cancer Institute for providing equipment for this work.

Additional Information and Declarations

Competing Interests

Author Contributions

Data Availability

The authors declare there are no competing interests.

Yu Du conceived and designed the experiments, analyzed the data, prepared figures and/or tables, and approved the final draft.

Heng Li analyzed the data, authored or reviewed drafts of the article, and approved the final draft.

Yan Wang performed the experiments, prepared figures and/or tables, and approved the final draft.

Yunyan He performed the experiments, prepared figures and/or tables, and approved the final draft.

Gaofeng Li conceived and designed the experiments, analyzed the data, authored or reviewed drafts of the article, and approved the final draft.

The following information was supplied regarding data availability:

The raw data are available in the Supplemental Files.

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
