# Peer review of "DLX1 acts as a novel prognostic biomarker involved in immune cell infiltration and tumor progression in lung adenocarcinoma"

_PeerJ, doi:10.7717/peerj.16823_

## Round 0.1 · original submission · Major Revisions

The manuscript needs major revision as indicated by the reviewers. Please address all the issues raised and resolve them in an exhaustive manner. Where additional experiments have been indicated they should be performed.

**Language Note:** The review process has identified that the English language must be improved. PeerJ can provide language editing services - please contact us at copyediting@peerj.com for pricing (be sure to provide your manuscript number and title). Alternatively, you should make your own arrangements to improve the language quality and provide details in your response letter. – PeerJ Staff

Reviewer 1 ·

Basic reporting

The manuscript by Du et al on DLX1 is largely based on publically available RNA expression databases with a focus on lung adenocarcinoimas. Very little bench laboratory experiments on a small number of lung cancer cell lines are presented. No in vivo experiments with hypothesis testing were performed. No corresponding protein expression is presented. The siRNA experiments do not control for off-target effects. Thus, the work, although not without merit, is reduced significantly in priority. While generally well written, multiple punctuation errors and the repeated lack of proper capitalization require correction. The list of differentially expressed genes in Fig 4B is not readable due to low resolution. In the introduction, it states that adenocarcinoma is the fastest metastatic and most aggressive type of lung cancer. However, this distinction is the property of SCLC.

DLX1, a homeodomain protein, is one of 6 family members. Other family members have been explored in different cancers, so it is unclear why only DLX1 was studied in lung adenocarcinoma. Among the 6 DLX genes, is DLX1 best correlated with outcome and, for example, what are the expression levels of other family members. It’s also of interest that squamous lung cancers (Fig 1A) express much higher (RNA) levels of DLX1, raising the question of why adenocarcinoma was the only focus of this study.

DLX function has been closely linked to TGFb and EMT. In leukemia, for instance, DLX1 and 2 expression is driven by growth factor signaling in an ERK and JNK-dependent manner, and was shown to inhibit nuclear phospho-SMAD2 levels and TGFb-mediated growth inhibition (Starkova et al., 2011). In murine breast cells, TGFb upregulated Dlx2 via upregulation of an EGFR ligand, betacellulin, which is also expressed in damaged bronchial epithelia and lung cancer cell lines (reviewed in Drabkin et al., 2017). This upregulation of Dlx2 was protective against TGFb-mediated growth inhibition and apoptosis. In a series of human breast cancers, DLX2 overexpression was mutually exclusive with p53 mutations. Whether this relationship with one or more DLX family members is present in lung adenocarcinoma is unknown. These questions are relevant to the current manuscript since invasion (e.g., thru matrigel) and wound closure assays are closely associated with EMT. Thus, growth factor signaling, especially involving EGFR and its ligands, the RAS-RAF-MAPK-ERK-JNK pathways, TGFb signaling and which EMT transcription factors, e.g., ZEB, Twist, etc., are most involved in the reported association with patient outcome are all outstanding questions. The multiple database analyses reported here are best used as hypothesis generating. Direct experimentation is required to test these hypotheses and to deal with the important issue of context specificity, where the downstream effects of a particular factor / pathway can be quite varied in different cell types. The above discussion is not meant to suggest that all questions should be addressed. Rather, the authors should be encouraged to focus on one or more of these or other areas and expand their studies in a hypothesis-based manner with substantial experimentation to confirm their findings.

Experimental design

see above
weak experimentally except for public database analysis

Validity of the findings

see above
insufficient experimentation to explore and validate hypotheses

Additional comments

See above. Low priority manuscript. Authors should be encouraged to do considerably more experimentation in a hypothesis-oriented manner. In its current form, I reject it completely. In the absence of major bench research experimentation, I would not re-review it. However, I will list it as requiring major revision.

·

Basic reporting

The use of English language is adequate and the manuscript flows well. There are appropriate references and background to justify the study rationale. The article structure, figures and tables are professional. The figures are self contained with relevant results to hypotheses.

A few comments:
1. Figure legends need more detail.
2. Figure 1D and E: is this redundant with the information from TCGA LUAD in figure 1B and C?
3. Figure 4A: should indicate the names of a few genes that are the most significantly associated with DLX1, either positively or negatively.
4. Figure 4B: the names of the genes are not legible in the pdf version, will need better resolution.
5. Figure 7F legend: Mast cells instead of Master cells
6. In Figure 9, the legend states that DLX1 methylation level is associated with prognosis, however this is not what the illustration shows.

Experimental design

The research question is within the Aims and Scope of the journal, research question is well defined, relevant and meaningful. Investigation is rigorous and methods are well described.

Validity of the findings

Underlying data are available in public databases. Additional data from cell line experiments and secondary analysis in the TCGA is provided in the supplement.

The authors overstate the conclusions of the study in the abstract (conclusion, lines 31-35, the authors state that DLX1 contributes to an immunosuppressive microenvironment and that is a target for treatment) and in the Discussion (conclusions: e.g. "Through the expression of genes involved in cell cycle regulation and immune response, DLX1 influences the progression of LUAD"). The authors have rather shown that there is an association of DLX1 expression with adverse outcomes and an immunosuppressive TME in LUAD. The in vitro work is focused on a mechanism different from immune escape (proliferation, migration, invasion).

In order to better support the conclusions, I suggest to perform RNAseq and methylation analysis in the A549 and H1299 cells with and without DLX1 knock downs. Then look at wether findings from GO, KEGG analysis and methylation before and after DLX1 knockdown explain what is seen in the TCGA data.

Reviewer 3 ·

Basic reporting

In the manuscript titled DLX1 acts as a novel prognostic biomarker involved in immune cell infiltration and tumor progression in lung adenocarcinoma, Yu Du et al. investigated the bioinformatics function and prognostic value of DLX1 involved in lung adenocarcinoma. This study contains some interesting findings and are valuable for the understanding that DLX1 is a prognostic biomarker and immunotherapy target for lung adenocarcinoma. However, lack of additional clinical data is the major flaw of the study. Therefore, minor revision has to be done before this manuscript could be accepted for publication in PeerJ.

Experimental design

The subject is designed scrupulously

Validity of the findings

The data need to be analyzed in more detail.

Additional comments

1. The current manuscript needs to be polished by a fluent English speaker or a professional language editing service.
2. More references are suggested.
3. The current manuscript needs to be polished.
4. Aspects of the analysis of DLX1's effect on lung adenocarcinoma were insufficient.

---

## Round 0.2 · Major Revisions

The manuscript still needs major revision before publication. Please follow the indications given by the reviewers. In particular, revise statements on diagnostic and prognostic value of DLX-1 expression either by removing or by substantiating it and add transcriptomic data on DLX-1 siRNA transfected cells in comparison to scrmabled siRNA transfected cells.

·

Basic reporting

See original review

Experimental design

No clear relevance. See other comments.

Validity of the findings

Conclusion about DLX1 as a therapeutic target in lung adenocarcinoma is not well supported by the results. See other comments.

Additional comments

The authors show an association of DLX1 expression with worse outcomes, less immune cell infiltration and processes like cell cycle progression and others in lung adenocarcinoma. Although these findings are hypothesis generating to suggest DLX1 as a therapeutic target, further in vitro experimentation on the involved mechanism is required. E.g. show the effects of DLX1 knockdown and overexpression in the transcriptome of available lung cancer cell lines. DLX1 as a diagnostic or prognostic biomarker does not have clear relevance on its own.

Reviewer 3 ·

Basic reporting

The discussion on functional enrichment analysis and the relationship between DLX1 expression and tumor immune cell infiltration as a whole stays on the description of results rather than further analysis.

Experimental design

1. The experimental verification part can be put at the end of the paper.

Validity of the findings

1. Figure 12: Calibration curves respond to the poor predictability of the model, whether to consider expanding the discussion in the article.

Additional comments

1. Line 43-44,” the most common type of lung cancer is lung adenocarcinoma (LUAD), which is the most aggressive and fastest metastatic type of lung cancer”, this is the property of SCLC.
2. Combining the paragraph “DEGs between LUAD patients with high and low expression of DLX1” and paragraph “Functional enrichment analysis of DLX1-related DEGs in LUAD” into one.

---

## Round 0.3 · accepted · Accept

Although there remain questions about the role of DLX1 on the tumor microenvironment, the paper, with the integrations included following the referees' indications, has some merit and can be considered suitable for publication.

Please consider the additional comments of R3.

·

Basic reporting

see previous review

Experimental design

see previous review

Validity of the findings

see previous review

Reviewer 3 ·

Basic reporting

The awareness that DLX1 is a predictive biomarker and therapeutic target for lung cancer is enhanced by the study's intriguing findings.

Experimental design

We hope that the authors will carry out corresponding in vivo and in vitro experiments in the future.

Validity of the findings

no comment

Additional comments

1.The background of the abstract part is better not to write DLX1 at the beginning, but to introduce DLX1 in a sentence.
2.SCLC is one of the most aggressive and rapidly metastatic types of lung cancer. The reference cited by the authors refers to the fact that lung ADC is still one of the most aggressive and rapidly fatal tumor types with overall survival less than 5 years.This sentence does not represent the properties of LUAD in lung cancer and can be changed to in tumors.